# Study on Factors Influencing Public Participation in River and Lake Governance in the Context of the River Chief System—Based on the Integrated Model of TPB-NAM

**Xia Zhang [1], Liqun Li [2], Zhaoxian Su [3],\*[ID], Haohao Li [4] and Xin Luo [3]**

1   College of Henan River Chief, North China University of Water Resources and Electric Power, Zhengzhou 450046, China
2   China South-to-North Water Diversion Middle Route Corporation Limited, Beijing 100045, China
3   School of Public Management, North China University of Water Resources and Electric Power, Zhengzhou 450046, China
4   Luoyang Branch of Henan Tobacco Company, Luoyang 471012, China
\*   Correspondence: suzhaoxian18@163.com

**Abstract:** Public participation in the context of the river chief system is not only beneficial for long-term river and lake governance (RLG), but it also is an important supplement to the existing governance mode led by the government. On the basis of the integrated model of TPB-NAM, this paper discusses the influencing factors and driving mechanisms of public participation in river and lake governance in the context of the river chief system from aspects of self-interest and altruism. Through the investigation of residents, 508 sample data points were obtained and analyzed by the structural equation model (SEM). The results were as follows: (1) the explanatory power of the TPB-NAM model ($R^2$ = 60.7%) was superior to that of the extended TPB model ($R^2$ = 60.0%) and NAM model ($R^2$ = 50.0%). (2) From the perspective of individual rationality, the intention for public participation in RLG had a significant positive influence on behavior, and behavior attitudes and subjective norms could predict their intentions. However, the roles were different; from the perspective of social rationality, awareness of negative consequences could actively affect positive personal norms through the ascription of responsibility, and positive personal norms could promote public participation in RLG. (3) Government norms were another important factor driving public participating in RLG. The results are of great theoretical significance for further exploring the public intention and behavior related to participation in RLG.

**Keywords:** river and lake governance; factors influencing public participation; SEM; integrated model of TPB-NAM; river chief system

## 1. Introduction

Rivers and lakes are important carriers of water resources, but they have been governed separately, and policies related to RLG have come from different departments in China, which has been difficult to change [1]. In order to solve the complex dilemma of RLG, local governments in China have explored effective practical strategies for water pollution control, and the river chief system (RCS) arose under this background [2]. The RCS is an important institutional guarantee to ensure the health of rivers and lakes. It originated 2007 when the Wuxi Municipal government entrusted the responsibility of RLG to the leaders of the Party and relevant government sectors at all levels in order to cope with serious water pollution in Taihu Lake Basin. The RCS achieved remarkable results by adopting mandatory policies and measures at the initial stage of RLG. However, long-term reliance on "river chief" administrative management has led to problems such as the single channel, lack of flexibility, and others. At present, RLG needs the joint participation of the public [3,4]. Although the relevant sectors of Chinese government have increasingly





strengthened public participation in the field of RLG, they fail to completely understand residents' intention and demands for participation, resulting in the current status quo of low public participation in RLG, incomplete understanding of policies, a weak sense of participation, and more [5,6].

Whether the public participates in RLG is a process of psychological decision making. The existing literature shows that the logical starting point for the study of this psychological decision-making process lies in the study of participation intention [7], while some scholars claim that the public tends to deviate from their behavior in the process of participation, so it is necessary to further study public participation behavior [8]. Existing studies have mainly focused on management of interest conflicts [9], integrated management mechanisms [10], non-political strategies [11], the institutional environment [12], the legal system framework [13], and other aspects, which mainly aim to explore how to make the public participate in RLG more effectively. However, at present, few scholars have discussed the psychological motivation and influencing factors of public participation in RLG in the current political and cultural context of the RCS in China. Public participation in RLG in the RCS context is essentially pro-environmental behavior to reduce environmental damage, with a mixed attribute of self-interest and altruism [14]. On the one hand, public participation is a kind of self-interest behavior to realize personal demands from a rational point of view, such as enjoying a good environmental ecology and ensuring their own health. On the other hand, public participation in RLG also includes the moral obligation of being responsible for the living environment and physical health of others or human beings to a certain extent [15,16].

Therefore, the paper integrates the theory of planned behavior (TPB) and the norm activation model (NAM), which are used to explain self-interest and altruism of public participation intention, respectively. Considering that SEM can deal with multiple dependent variables at the same time and has the advantages of estimating the factor structure and relationship simultaneously as well as estimating the fitting degree of the whole model, it meets the needs of this study well. By applying SEM, the paper verifies the explanatory power of the theoretical model through empirical methods and thoroughly discusses the influencing factors and driving mechanisms of public participation in RLG in the RCS context, which aims at providing countermeasures and scientific paths for stimulating the public participation in RLG. The major contribution of this paper is that it combines TPB with NAM and applies the integrated model of TPB-NAM to the study of public participation in RLG in the RCS context, which provides a new perspective for the study of public pro-environmental behavior.

The rest of this paper is organized as follows. Section 2 introduces the extended TPB model, NAM, the integrated model of TPB-NAM, and the relevant research hypotheses. Section 3 presents the scale design and sample selection. Section 4 provides the results of the three models and discusses them. Section 5 summarizes the study conclusions and proposes corresponding suggestions.

## 2. The Theoretical Framework and Research Hypothesis

### 2.1. The Extended TPB Model

The TPB was proposed and improved by Ajzen. It is an extension and expansion of rational behavior theory. At present, it has become one of the models which are most common used to predict individual behavior. The theory holds the view that intention is the most direct factor influencing behavior and that behavior attitudes, subjective norms, and perceived behavior control are the antecedents of behavior intention [17,18].

The behavior attitude (BA) is the degree of positive or negative evaluation that an individual holds for the implementation of a specific behavior. BA puts more emphasis on the overall expected evaluation of individual behavior according to the actual situation, which is actually a kind of preference for behavior [19]. Generally speaking, when individuals hold a more active BA towards a certain behavior, they are more willing to engage in this behavior. For example, some studies revealed that attitudes towards the

environment would encourage individuals to express their intention to act in a way that was responsible for themselves and the environment, especially in the case of incomplete information on environmental public participation in decision making and unequal status among participants [20]. In research on the integrated collective action from individual rational action to social identity, Le et al. pointed out that BA would have an impact on behavior intention [21]. If the public thinks that their own participation in RLG can improve water quality, promote the surrounding vegetation in the environment, and even create economic value, they will stimulate their own interest in participation and show a strong intention to participate. Therefore, the hypothesis regarding BA is:

**Hypothesis 1 (H1).** *A positive BA of the public can actively affect their intention to participate in RLG.*

The subjective norm (SN) refers to the social pressure that individuals feel when deciding whether to implement some behavior. In the TPB, it is generally accepted that the stronger the subject's cognition of the SN is, the stronger his intention to act will be. Individuals usually tend to be consistent with expectations or behaviors of the reference group, so the recognition of individual behavior by the reference group will drive the intention of individuals to implement the behavior. For example, in the research on the factors influencing social stability risk assessment, Zhu et al. verified that SN had a direct positive impact on public participation intention in social risk assessment [22]. When deciding whether to protect the ecological environment, the public tends to be influenced by surrounding people or organizations such as family, relatives, friends, neighbors, and government sectors. The stronger the perceived social pressure of the government and others is, the more likely it is that the public will be motivated to participate in RLG. Therefore, the hypothesis regarding SN is:

**Hypothesis 2 (H2).** *Positive SN of the public can positively affect their participation intention in RLG.*

Regarding participation intention (PI), the core view of TPB is that positive psychological factors of the subjects will strengthen their positive intention to make behavioral choices, and this positive intention will inevitably lead to positive behavioral choices [23]. In this study, this means that near rivers and lakes, the more active the public's attitude towards RLG is and the greater the demonstrated influence of others or organizations around them is, the stronger their PI will be. Therefore, the hypothesis regarding the intention is:

**Hypothesis 3 (H3).** *The active PI of the public can positively affect their participation in RLG.*

Government norms (GN) can also influence public participation. The nature of public participation in environmental governance is to transform the discretion of the government at the implementation level into the active participation of the public. An open government will improve the participation rate of citizens. Openness refers to the open, responsive, and inclusive attitude of the government towards public participation, which is a cornerstone of government credibility, i.e., government behavior norms [24]. For example, Ying et al.believed that the impact of public participation in environmental governance was highly reliant upon the central government and that public environmental participation could productively curb the negative impact of improper local government intervention on air quality [25]. In addition, Yan et al. claimed that the level of public participation in water pollution control would be affected by the behavior of the government in the process of system operation [26]. Therefore, public participation in RLG is not only affected by the psychological factors proposed by the TPB but also by GN; GN can not only indirectly affect participation through PI but can also have a direct impact on participation. Consequently, the hypotheses regarding GN are:

**Hypothesis 4 (H4).** *Normative government behavior can positively affect the intention of the public to participate in RLG.*

**Hypothesis 5 (H5).** *Normative government behavior can positively affect the public's participation in RLG.*

*2.2. The Framework of NAM*

Although the TPB model is important for predicting individual behavior, it fails to account for the influence of altruistic motivation when analyzing individual pro-environmental behavior, resulting in the reduction in the explanatory force. Schwartz proposed NAM with personal norms as the core factor, which is mainly used to predict and understand pro-social behavior and altruistic behavior. This model is composed of personal norms (PN), ascription of responsibility (AR), and awareness of consequences (AC). According to NAM, the motivation for individuals to implement pro-environmental behavior comes from their internal sense of moral obligation, that is, PN, which is activated by both their AC and AR; AR is the intermediary variable of the relationship between AC and PN, and PN is the intermediary variable of responsibility and behavior [27,28]. Activation of the PN of the public is a prerequisite for their participation in RLG. Only when the public is more clearly conscious of adverse consequences of not participating in RLG and more likely to attribute responsibility to themselves can they have stronger PN of their participation in RLG. After the public's PN is activated, their non-participation in RLG may cause a strong sense of guilt and self-blame, and such sense of moral obligation could drive them to participate in RLG.

Personal norms (PN) are derived from a strong internal sense of obligation to implement pro-environmental behavior, which can also be called moral norms. For example, Wan et al. revealed that moral factors with personal norms as the core had a significant role in promoting public PI in research on public participation in nearby construction projects [29]. Similarly, by analyzing the factors that influenced urban residents' participation in environmental governance, Wang and Zhang demonstrated that personal norms had the greatest direct impact on urban residents' participation in environmental governance, and individual moral responsibility played an important role in the formation of urban residents' intention to participate in environmental governance [30]. As for public participation in RLG, PN means that the public regards participation in RLG as their own norms and moral obligations, and not participating in RLG will produce a series of emotions such as guilt. Therefore, the hypothesis regarding PN is:

**Hypothesis 6 (H6).** *Public PN can positively affect their participation in RLG.*

Ascription of responsibility (AR) is the individual's sense of responsibility for adverse consequences resulting from not implementing a certain behavior. When individuals are conscious of the negative results of their behavior on others, they tend to attribute responsibility for the consequences to themselves [31]. Through empirical analysis, Ge et al. stated that the public's awareness of environmental responsibility could promote their participation in environmental protection and that individuals with a high sense of environmental responsibility not only considered their own economic interests but also considered the impact of behavior on the environment so as to implement more pro-environmental behaviors [32]. In terms of RLG, AR mainly refers to the awareness that the public bears a certain responsibility for the consequences, such as ecological damage, caused by not participating in RLG. Therefore, the hypothesis regarding AR is:

**Hypothesis 7 (H7).** *Public AR can positively affect PN in RLG.*

Awareness of consequences (AC) involves individuals' awareness of negative effects caused to others or other things by not implementing the target behavior, which represents

the individual's awareness that not implementing a certain behavior may bring adverse consequences to others [33]. Vanliere and Dunlap were the first to apply NAM to study environmental protection behavior; they studied courtyard waste combustion [34]. Their research indicated that the stronger the AC, the less likely residents were to engage in garbage combustion behavior. This conclusion is consistent with NAM's basic assumptions. As for public participation in RLG, if the public does not participate in RLG, they may recognize that it can cause serious negative consequences, such as the destruction of river and lake ecology, the decline of urban living environment quality, an adverse impact on residents' health, and the waste of many resources. Therefore, the higher the public's perception of serious consequences of not participating in RLG is, the greater the likelihood that people will participate in RLG. Consequently, the hypothesis regarding AC is:

**Hypothesis 8 (H8).** *Public AC can positively affect the responsibility of RLG.*

### 2.3. The Integrated Model of TPB-NAM

On the one hand, the TPB takes the self-interest of individuals as the starting point and holds the view that individuals make choices based on the rational choice of maximizing net income; that is, individuals decide whether to engage in a certain behavior after measuring the cost and income, ignoring the role of irrational and altruistic motivation in shaping behavior. This theory is not adequate to effectively explain the emergence of pro-environmental behavior. On the other hand, as a rational choice model, NAM assumes that individuals' implementation of some behavior rests upon the thought of moral obligation and emphasizes the importance of morality, which makes NAM lack explanatory power in terms of self-interest-driven behavior [35,36]. Considering the fact that river and lake environmental governance behavior in the RCS is a public behavior that involves both self-interest and altruism, studying the psychological factors that influence residents' participation in RLG from the perspective of self-interest alone will inevitably neglect the public welfare aspect of the environmental behavior, and the psychology of individual altruism also affects individual environmental choices in a way that is not perceived by the main body [37]. Therefore, this study integrates the TPB model with NAM and analyzes the influence of psychological factors and government behavioral norms on the public participation in RLG in the RCS, with the aim of developing explanatory power of the theory on the intention or behavior of public participation in RLG (as is shown in Figure 1).

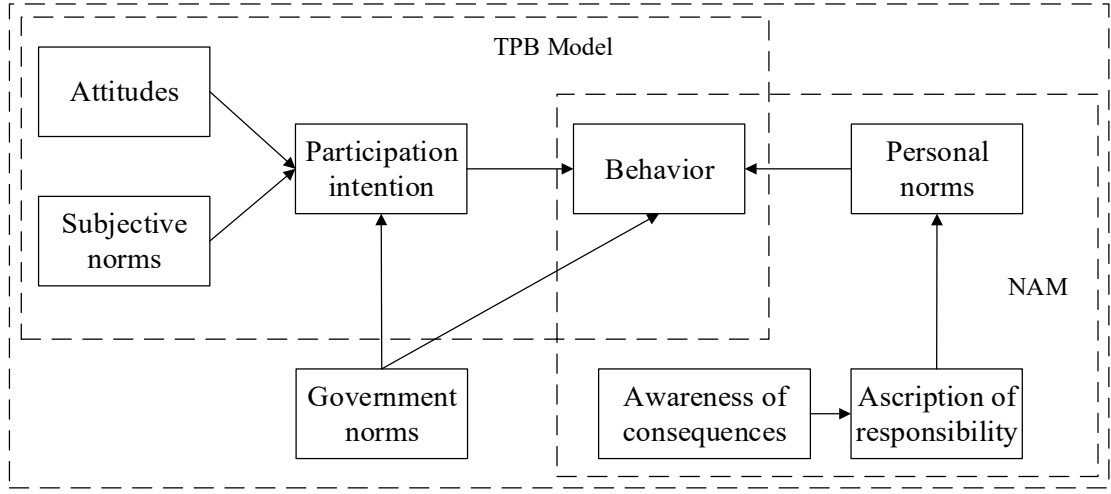

**Figure 1.** The integrated model of TPB-NAM.

## 3. Materials and Methods

### 3.1. Scale Design

On the basis of theoretical analysis, referring to the index selection and scale design results of relevant research [6,9,14,26,29,30], combined with the current actual public participation in RLG in the context of the RCS, we designed 26 items. Using a five-level Likert scale (from one being "strongly disagree" to five being "strongly agree"), we measured eight variables, which were participation behavior (PB), behavior attitude (BA), subjective norms (SN), participation intention (PI), government norms (GN), personal norms (PN), awareness of consequences (AC), and ascription of responsibility (AR) (as is shown in Table 1).

**Table 1.** Questionnaire on public participation in RLG (*n* = 508).

| Construct | Number | Observable Variables | Items | Mean Value | Standard Deviation |
|---|---|---|---|---|---|
| Attitudes | BA1 | ecological rationality | I think public participation in RLG can effectively improve the water quality of rivers and lakes. | 3.56 | 0.975 |
| | BA2 | ecological rationality | I think public participation in RLG can effectively improve the surrounding vegetation environment. | 3.94 | 0.988 |
| | BA3 | | I think public participation in RLG can improve the surrounding sanitary conditions. | 3.50 | 1.202 |
| | BA4 | economic rationality | I think public participation in RLG can create economic benefits for ecological civilization. | 3.59 | 0.930 |
| Subjective norms | SN1 | indicative norms | The government encourages public participation in RLG. | 3.68 | 0.967 |
| | SN2 | indicative norms | The government encourages the public to serve as non-government river chiefs. | 4.14 | 0.911 |
| | SN3 | descriptive norm | People around generally have good ecological environment protection behavior. | 3.66 | 0.939 |
| | SN4 | descriptive norm | The behavior of polluting rivers and lakes will be morally condemned, supervised and reported by the public. | 3.37 | 0.983 |
| Participation intention | PN1 | attention intention | I will obtain relevant RLG information through formal and informal channels such as the Internet and government release. | 4.03 | 0.970 |
| | PN2 | propaganda intention | I will participate in publicity and education activities related to RLG. | 4.01 | 0.967 |
| | PN3 | governance intention | When the rivers in the community are damaged, I am willing to participate in RLG. | 3.52 | 1.037 |
| Participation behavior | PB1 | supervising enterprises' behavior | When enterprises cause pollution, I will supervise and hold them accountable | 3.62 | 0.928 |
| | PB2 | supervising others' behavior | I will prevent others from destroying the river environment or littering. | 3.46 | 1.047 |
| | PB3 | reporting acts | I will report the destruction of rivers and lakes. | 3.95 | 0.937 |
| Government norms | GN1 | attention degree | The government will attach importance to the opinions and suggestions put forward by the public in the process of participating in RLG. | 3.56 | 0.960 |
| | GN2 | responsiveness | The government will respond to the public's monitoring and reporting of river and lake pollution. | 4.00 | 0.930 |
| | GN3 | enforceability | The government will actively deal with the illegal sewage discharge behavior of enterprises reported by the public. | 3.59 | 0.936 |
| Awareness of consequences | AC1 | ecological environment | The public's non-participation in RLG may lead to the decline of the surrounding ecological environment. | 3.64 | 0.923 |
| | AC2 | quality of life | Public participation in RLG can improve the living environment and health of the public. | 3.64 | 0.923 |
| | AC3 | personal habit | Public participation in RLG can help people develop awareness and habits of caring for the environment | 3.40 | 1.043 |
| Ascription of responsibility Personal norms | AR1 | pollution hazard | In order to reduce the damage caused by river and lake pollution, I have the responsibility to participate in RCS to control rivers and lakes pollution. | 3.64 | 0.967 |
| | AR2 | ecological construction constraints | I have a certain responsibility for the constraints on the construction of ecological civilization caused by not participating in RLG. | 3.70 | 1.015 |
| | AR3 | policy response | The relevant government departments should be responsible for RLG. | 3.58 | 0.964 |
| | PN1 | moral principle | Participating in RLG conforms to my moral principles and values. | 3.66 | 0.983 |
| | PN2 | moral duty | Not participating in RLG would make me feel guilty. | 3.36 | 1.064 |
| | PN3 | | Participating in RLG would make me feel satisfied. | 3.96 | 0.940 |

### 3.2. Data Sources and Sample Descriptions

Entrusted by the River and Lake Protection Center of Ministry of Water Resources of China, we used the residents along the river and lake in the area under the jurisdiction of Henan Province as the research subjects and randomly distributed 540 questionnaires

entitled *Questionnaire on public participation in RLG* in 20 counties (districts) within the region. After removing the invalid questionnaires, the final number of the questionnaires collected was 508, and the final effective proportion of the questionnaire was 94.07%. The survey was conducted in 18 cities under the jurisdiction of Henan Province. These 18 cities served as sample areas, three or four towns were randomly selected according to the population and economic conditions of each sample city, and three or four villages were randomly selected from each township, which fully reflected the representativeness of the samples in terms of geographical distribution and socio-economic development level. Before issuing the questionnaire, we first interviewed experts in the field and conducted 30 pre-surveys. The specialists include 12 experts and scholars, who are mainly engaged in RLG and the RCS in universities and scientific research institutes, and 8 government staff, who are mainly working staff of the River Lake Protection Center of the Ministry of Water Resources of China. After modifying the questionnaire items and language expressions, the final version was obtained. The formal survey of this study was carried out from 1 February 2021 to 31 September 2021. In order to ensure the quality of data, each questionnaire was completed by trained members of the research group through one-on-one interviews (empirical data were collected through structured questionnaires).

The distribution of this questionnaire was relatively reasonable, and the age, education background, and income level of the respondents were relatively consistent with the reality (the specific characteristic information is shown in Table 2). The age of respondents mainly ranged from 30 to 49, and the study included 298 males and 210 females. The respondents' main academic qualifications included the level of university, high school, and below, and their annual family income was mainly concentrated in the range of 20,000–60,000 Yuan.

**Table 2.** Demographic characteristics of samples.

| Basic Features | Category Descriptions | Sample Number | Proportion |
| --- | --- | --- | --- |
| Age | 21~29 | 91 | 17.91% |
| | 30~39 | 164 | 32.28% |
| | 40~49 | 139 | 27.36% |
| | 50~59 | 87 | 17.13% |
| | Over 60 | 27 | 5.32% |
| Gender | Male | 298 | 58.66% |
| | Female | 210 | 41.34% |
| Academic level | Doctor | 3 | 0.59% |
| | Master | 33 | 6.50% |
| | University | 111 | 21.85% |
| | High/secondary/technical School | 203 | 39.96% |
| | Under senior high school | 158 | 31.10% |
| Family annual income | Less than 20,000 Yuan | 83 | 16.34% |
| | 20,000–60,000 Yuan. | 227 | 44.69% |
| | 60,000–100,000 Yuan | 142 | 27.95% |
| | More than 100,000 Yuan | 56 | 11.02% |

### 3.3. Methods

SPSS 24.0 and AMOS 24.0 were applied for quantitative analysis of the data. First, reliability and validity were measured, and confirmatory factor analysis (CFA) was conducted on the survey data to evaluate the goodness-of-fit of the models. Then the hypotheses of the structural model were verified. Structural equation models can not only analyze the interaction between multiple latent variables but also avoid the collinearity problem in multiple regression. The specific model and equation are as follows:

$$X = \Lambda_X \xi + \delta \tag{1}$$

$$Y = \Lambda_X \eta + \varepsilon \tag{2}$$

$$\eta = \beta\eta + \Gamma\xi + \varsigma \tag{3}$$

Specifically, we conducted an empirical evaluation of Cronbach's $\alpha$, KMO value, composite reliability, and structural validity. It is generally believed that if the Cronbach coefficient is greater than 0.7, the reliability is higher. Convergence validity requires that the standardized factor load estimation (Std.) and average variance extraction value (AVE) should be greater than 0.5, and the constituent reliability (CR) should be greater than 0.7.

### 3.4. Reliability and Validity Tests

SPSS 24.0 was used to analyze the data. The results showed that the Cronbach coefficient of the whole scale and each latent variable was between 0.743–0.819, which was greater than the threshold condition of 0.7. KMO values of all latent variables were between 0.683–0.801, which was greater than the threshold condition of 0.5. Additionally, the adjoint probability of the Bartlett sphericity test was less than 0.001. AMOS 24.0 was applied to conduct CFA on the measurement model; the CR value was basically consistent with the Cronbach coefficient, and the scale passed the reliability test. The standardized factor load of variables was basically above 0.7, and $p < 0.001$, which was statistically significant. In addition, AVE of all latent variables was greater than 0.5, and the model scale had good convergent validity (as is shown in Table 3).

**Table 3.** Analysis results of variable reliability, validity, and factors.

| Items | Std | Reliability Test | | | Validity Test | AVE |
| | | Cronbach's $\alpha$ | CR | KMO | Bartlett Sphericity Test | |
|---|---|---|---|---|---|---|
| BA1 | 0.79 | | | | | |
| BA2 | 0.76 | 0.819 | 0.8274 | 0.801 | 711.757 ($p < 0.001$) | 0.5462 |
| BA3 | 0.66 | | | | | |
| BA4 | 0.74 | | | | | |
| SN1 | 0.69 | | | | | |
| SN2 | 0.77 | 0.809 | 0.8334 | 0.799 | 639.137 ($p < 0.001$) | 0.5561 |
| SN3 | 0.76 | | | | | |
| SN4 | 0.76 | | | | | |
| PI1 | 0.78 | | | | | |
| PI2 | 0.75 | 0.785 | 0.7878 | 0.702 | 437.121 ($p < 0.001$) | 0.5536 |
| PI3 | 0.70 | | | | | |
| PB1 | 0.72 | | | | | |
| PB2 | 0.71 | 0.748 | 0.7531 | 0.687 | 352.494 ($p < 0.001$) | 0.5042 |
| PB3 | 0.70 | | | | | |
| GN1 | 0.73 | | | | | |
| GN2 | 0.85 | 0.804 | 0.8056 | 0.701 | 495.475 ($p < 0.001$) | 0.5818 |
| GN3 | 0.70 | | | | | |
| AC1 | 0.69 | | | | | |
| AC2 | 0.83 | 0.763 | 0.7725 | 0.683 | 397.274 ($p < 0.001$) | 0.5335 |
| AC3 | 0.66 | | | | | |
| AR1 | 0.69 | | | | | |
| AR2 | 0.83 | 0.791 | 0.7954 | 0.700 | 455.969 ($p < 0.001$) | 0.5660 |
| AR3 | 0.73 | | | | | |
| PN1 | 0.81 | | | | | |
| PN2 | 0.70 | 0.789 | 0.7916 | 0.702 | 450.944 ($p < 0.001$) | 0.5597 |
| PN3 | 0.73 | | | | | |

## 4. Results and Discussion

### 4.1. Results

Three competing models were constructed to determine the most suitable model for the mechanisms impacting public participation in RLG. These are competing model 1 (TPB model), competing model 2 (NAM) and competing model 3 (the integrated model of TPB-NAM). In competing model 1, the factors influencing public participation in RLG were mainly explored by the extended TPB model with government behavior norms. In competing model 2, the mechanisms influencing public participation in RLG were mainly explored through the initial NAM. In competing model 3, the mechanisms influencing pub-

lic participation in RLG were mainly explored by integrating the extended TPB model and the initial NAM. Table 4 demonstrates the path coefficients of the three competing models.

**Table 4.** The estimated results of three competing models.

| Path | Competing Model 1: TPB Model | Competing Model 2: NAM | Competing Model 3: Integrated Model of TPB-NAM |
|---|---|---|---|
| BA → PI | 0.298 *** | | 0.288 *** |
| SN → PI | 0.622 *** | | 0.627 *** |
| PI → PB | 0.355 *** | | 0.270 *** |
| GN → PI | −0.064 n's | | −0.055 n's |
| GN → PB | 0.503 *** | | 0.405 *** |
| AC → AR | | 0.802 *** | 0.919 *** |
| AR → PN | | 0.838 *** | 0.821 *** |
| PN → PB | | 0.708 *** | 0.235 *** |
| Goodness-of-fit index | CMIN/DF = 1.757 ($p$ = 0.00); GFI = 0.953; IFI = 0.977; NFI = 0.949; CFI = 0.977; RMSEA = 0.039 | CMIN/DF = 2.789 ($p$ = 0.00); GFI = 0.956; IFI = 0.962; NFI = 0.942; CFI = 0.962; RMSEA = 0.059 | CMIN/DF = 1.855 ($p$ = 0.00); GFI = 0.919; IFI = 0.960; NFI = 0.917; CFI = 0.960; RMSEA = 0.041 |
| Goodness-of-fit index for model comparison | CMIN = 195.072 AIC = 279.072 BIC = 456.753 ECVI = 0.55 | CMIN = 142.237 AIC = 196.237 BIC = 310.460 ECVI = 0.387 | CMIN = 528.593 AIC = 660.593 BIC = 939.805 ECVI = 1.303 |
| Explanatory power | 0.600 | 0.501 | 0.607 |

Note: *** is significant at 1%. ns means not significant. Robust standard error in parentheses.

### 4.1.1. Competing Model 1

Regarding competing model 1, Figure 2 demonstrates the estimated model (TPB model) with standardized path coefficients and the overall fit index of the model $\chi2/df$ = 1.757 (less than 5), RMSEA = 0.039 (less than 0.08), GFI = 0.953 (more than 0.9), IFI = 0.977 (more than 0.9), NFI = 0.949 (more than 0.9), and CFI = 0.977 (more than 0.9). All fitting indicators met the threshold conditions. The results of the fitness test showed that the proposed model fit well with the actual survey data and had the analytical characteristics of the structural equation model. Competing model 1 jointly explained 60.0% of the variance in participation behavior ($R^2$ = 0.600).

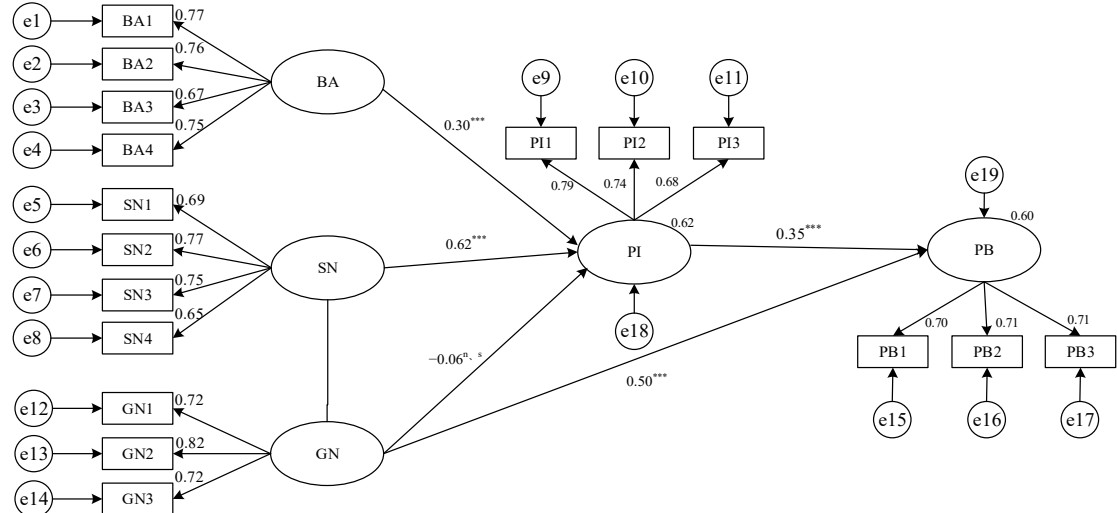

**Figure 2.** Competing model 1: TPB model. Note: *** is significant at 1%. ns means not significant. Robust standard error in parentheses.

On the basis of the test results of SEM (Figure 2), we found that the public behavior attitude (β = 0.30, $p$ < 0.001) and subjective norms (β = 0.62, $p$ < 0.001) had a significant positive impact on the public PI. In other words, hypotheses one and two were verified. The intention of the public to participate in RLG could predict and explain their participation behavior to a large extent, while other factors only indirectly affected the public's

participation in RLG. Among them, SN was the strongest predictor of public PI, and BA was also an important influencing factor of public PI. Both SN and BA played a positive role in promoting public PI. SN had the strongest positive influence on public PI, and the degree of influence was the most obvious in the model. As most of the areas near rivers and lakes are in the suburbs of cities or villages and towns, many residents in the area are over middle-aged and have lower education and lack of understanding of participation in RLG. These characteristics were also confirmed by the survey subjects in this paper. Therefore, there is a general mindset about the effect of participation, and the PI is vulnerable to social pressure from family members, acquaintances, and so on. To some extent, this also confirms the view of Rhee et al [38]. that people from collectivist cultures have a more significant SN.

Public PI (β = 0.35, $p < 0.001$) and GN (β = 0.50, $p < 0.001$) had a significant positive impact on public participation in RLG, meaning hypotheses three and five were verified. In addition, the impact of GN on the intention of public participation was not significant. Hypothesis four failed to pass the test and had no statistical significance. A possible reason is that the intention of the public to participate in RLG lies more in their own interests and social responsibility. The GN had no direct relationship with individuals' PI in RLG.

### 4.1.2. Competing Model 2

Regarding competing model 2, Figure 3 displays the estimated model (NAM) with standardized path coefficients and the overall fit index of the model $\chi^2/\text{df} = 2.789$ (less than 5), RMSEA = 0.061 (less than 0.08), GFI = 0.956 (more than 0.9), IFI = 0.962 (more than 0.9), NFI = 0.942 (more than 0.9), and CFI = 0.962 (more than 0.9). All fitting indicators of the model met the threshold conditions. The estimated model was well adapted to the actual survey data and had the analytical characteristics of the structural equation model. Competing model 2 jointly accounted for 50.1% of the variance in participation behavior ($R^2 = 0.501$).

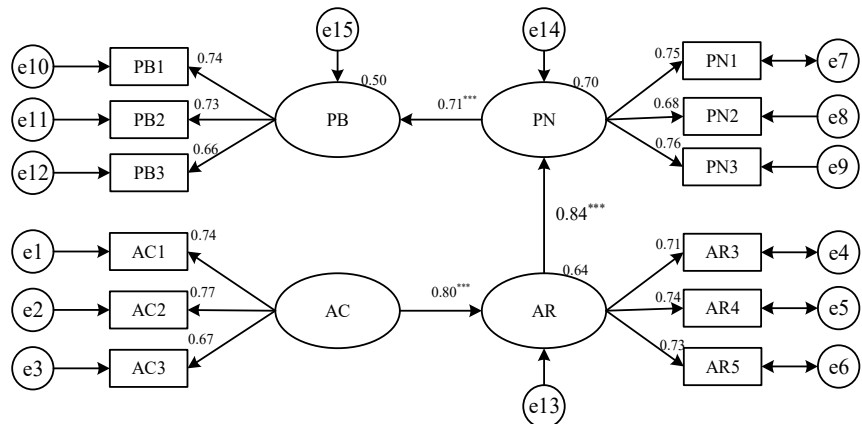

**Figure 3.** Competing model 2: NAM model. Note: *** is significant at 1%. Robust standard error in parentheses.

Hypotheses six, seven, and eight, which are related to the NAM, were all proven at the level of 0.001, meaning the PN of the public (β = 0.708, $p < 0.001$) significantly promoted their participation in RLG, which in turn means that if individuals believe they have a moral obligation to participate in RLG, they are more likely to implement the corresponding behavior. Public AR (β = 0.838, $p < 0.001$) stimulated their PN, and AC (β = 0.802, $p < 0.001$) indirectly had a significant impact on PN by affecting their AR. In other words, the more an individual feels that not actively participating in RLG brings adverse consequences, the more likely he is to actively participate in RLG. The sense of moral obligation generated by the public's self-attribution of the consequences of the effects of not participating is also a motivational factor affecting their participation in the decision making of RLG projects, which indicates that improvement of public awareness of environmental responsibility can

significantly promote the intention to participate in RLG. This is similar to the conclusion of the research of Enriquez-Acevedo [39]. The active response of the study subjects to the relevant issues in the interview also confirmed the role of moral factors in such public participation in RLG. The verification of moral motivation has important implications for relevant policy making.

### 4.1.3. Competing Model 3

Regarding competing model 3, the TPB model with the government behavior norms was integrated with the initial NAM to explore the mechanisms impacting public participation in RLG. Figure 4 demonstrates the estimated model (Integrated Model of TPB-NAM) with standardized path coefficients. The overall fit index of the model $\chi^2/df = 1.855$ (less than 5), RMSEA = 0.041 (less than 0.08), GFI = 0.919 (more than 0.9), IFI = 0.960 (more than 0.9), NFI = 0.917 (more than 0.9), and CFI = 0.960 (more than 0.9). Each fitting index of the model met the threshold conditions and passed the fitness test, which shows that the estimated model was well adapted to the actual survey data and had the analysis characteristics of the structural equation model. Competing model 3 explained a total of 60.7% of the variance in public participation in RLG ($R^2 = 0.607$). The explanatory power was higher than that of competing model 1 and competing model 2, so indicating that competing model 3 was better than the other two models.

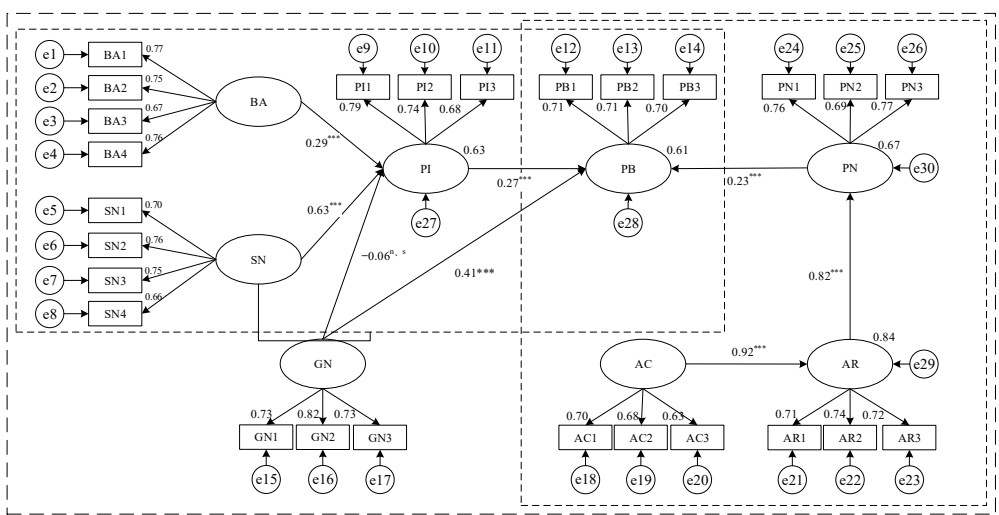

**Figure 4.** Competing model 3: the integrated model of TPB-NAM. Note: *** is significant at 1%. ns means not significant. Robust standard error in parentheses.

The path analysis results of the integrated model demonstrated that the relationship between attitudes, norms, and intentions was statistically significant (shown in Table 4), meaning hypotheses one and two were also verified in the integrated model. In terms of the factors influencing intention, the results were similar to the those in competing model 1, and the standardized path coefficient of SN was the largest ($\beta = 0.627$, $p < 0.001$), followed by behavior attitude ($\beta = 0.288$, $p < 0.001$). The results revealed that in the RCS context, SN has the largest direct impact on the public's intention to participate in RLG, which is consistent with the research conclusion of Wang [40]. This means that public participation in RLG is subject to social pressure from important people such as relatives, friends, and neighbors. The pressure has a positive impact on public participation in RLG, thus increasing their intention to participate. BA had a significant positive impact on the public's intention to participate in RLG and was the second major factor affecting intention. The research of Ma et al. [41] and Carmi [42] also confirmed this result. When the public feels that their participation in RLG can improve the surrounding ecological environment, their intention to participate is significantly increased. The impact of GN on the public PI was still not significant, so it is speculated that the GN level does not affect the public PI but significantly

affects the actual participation behavior. This is because PI is only a psychological tendency, and GN may not necessarily have an impact on the public's psychological intention.

In terms of the factors influencing participation behavior, GN ($\beta$ = 0.405, $p$ < 0.001), PI ($\beta$ = 0.270, $p$ < 0.001), and PN ($\beta$ = 0.235, $p$ < 0.001) had a significant positive impact on public participation in RLG, meaning hypotheses three, five, and six were also verified in the integrated model, in which GN had a greater impact on public participation. GN was the most important factor underlying public participation in environmental governance under the leadership of the government, which is more consistent with Hiromi's research [43]. In other words, GN directly incentivized the public to participate in environmental governance, and it was the key to increasing public participation in RLG. AC ($\beta$ = 0.919, $p$ < 0.001) positively affected the formation of positive public PN through AR ($\beta$ = 0.821, $p$ < 0.001), meaning hypotheses seven and eight were verified. The result is consistent with the research conclusion of Liu et al. [44]. If individuals understand the possible negative consequences of not participating in environmental governance and do not participate in RLG, it will be difficult for them to form high ethical standards. When the public clearly realize the importance of river and lake environmental governance, they will be encouraged to enhance their moral obligations and social responsibilities. According to Onwezen et al. [45], once the code of ethics is activated, people's social responsibility will be enhanced accordingly, and the relevant altruistic behavior will become their first choice. PN directly influences their actual participation behavior, and the stronger the PI is, the more likely they are to act.

In the investigation, we found that the public's PI in RLG was not consistent with the actual PB. On the one hand, even if individuals had high individual norms, the number of individuals participating in RLG was very small or even absent; on the other hand, from the perspective of the transformation of intention to behavior, the public PI in RLG was not effectively transformed into actual behavior, and there was a certain contradiction between them. The following factors may account for this phenomenon. First, some local river chiefs did not often disclose information related to the process of RLG and made direct decisions without soliciting public opinions, making public participation in RLG a mere formality. Second, some local river chiefs only solicited the opinions of experts and the public who were in line with their own intentions. Public groups had objections to the governance of rivers and lakes, but they had no opportunity to express them. Third, some local river chiefs did openly call on the public to participate in RLG, but due to the single method and narrow scope of the appeal, only a few people had the opportunity to participate in RLG. Therefore, even if the public had a strong intention to participate and strong personal norms, they did not participate.

*4.2. Discussion*

The purpose of this study was to explain the factors influencing public participation in RLG in the RCS context. Therefore, the paper introduced three different theoretical models to explain the behavior of public participation in RLG. The extended TPB model (model 1) and NAM (model 2) proved that public participation in RLG can be interpreted by self-interest and altruism. The results confirmed the effectiveness of the extended TPB model and NAM in explaining public participation in RLG. Theoretically, public participation in RLG in the RCS context is an environmentally friendly behavior. Whether the public is willing to participate in RLG is jointly driven by interest factors and moral factors. In terms of self-interest, public participation in RLG is influenced by the public's perception of BA, SN, and GN. Only when the public believes that participation in RLG can bring good results and meet the expectations of important individuals can they have a strong intention to participate. In terms of altruism, it is mainly determined by the level of their personally embedded moral standards. Whether morality is motivated depends on the public's understanding of the results of participation or non-participation and whether they will attribute these results to themselves. If the public realizes that participation in RLG has some benefits, such as improving the living environment and environmental health, and

believes that they should be responsible for the negative consequences of non-participation, it will generate strong intention to participate.

Therefore, the study took into account the egoistic and altruistic elements of public environmental participation and introduced psychological factors that affect public participation into the TPB and NAM (model 3). We first tested the extended TPB (model 1) and NAM (model 2). Then, the goodness-of-fit of the integrated model of TPB-NAM (model 3) was analyzed after all variables were added. The fitting indexes of the three models all attained acceptable fitting degrees, of which $R^2$ in the integrated model was greater than that of the first two models. The practicability and applicability of the integrated model was superior to the extended TPB model and NAM in predicting public participation in RLG in the RCS context, which is more consistent with the research conclusions of Kim et al. [46], Chun et al. [47], Shen et al. [48], Rezaei et al. [49], and Li et al. [50].

Public PB in RLG was influenced by the interaction of BA, SN, GN, PI, AR, PN, and AC. The study combined different influencing factors to strengthen the understanding of public participation in RLG in the RCS context. Through empirical analysis, the impact of the above factors on public participation in RLG was confirmed. First, GN and SN had the strongest positive influence on the public's PI. When the public's demands get immediate feedback from the government, more people pay attention to RLG, generating huge ecological and economic benefits; when the public is affected by social pressures from family members, acquaintances, etc., they will further generate strong intention to participate or even directly participate in RLG. Second, in addition to self-interest motivation, the public's perception of the results of participation and the sense of moral obligation arising from the self-attribution of the results are also motivational factors that affect their participation in RLG. The positive responses of study subjects to relevant questions in the interview also confirmed the role of moral factors in such public participation. Finally, the study revealed that the public's PI was not strong enough, that the intention to participate in RLG has not effectively translated into actual behavior, and that there is a certain contradiction between the two aspects. The public participation in environmental governance in China is still at the primary level. The solidification of the underlying logic of social governance and the strong dependence of the public have become important factors preventing the transformation of China's environmental governance model. In fact, the development and promotion of public participation in environmental governance is a long-term and slow process.

## 5. Conclusions

### 5.1. Summary

Considering the self-interest and altruism of public participation, the psychological factors affecting public participation were incorporated into the TPB and NAM framework to study public participation in RLG in the context of the RCS, which provides a new perspective for the study of public participation in environmental governance. The explanatory power of TPB-NAM model ($R^2$ = 60.7%) was superior to that of the extended TPB model ($R^2$ = 60.0%) and NAM model ($R^2$ = 50.0%). In terms of self-interest, it depends on the public's awareness of behavior attitude and subjective norms. When the public believes that participating in RLG can bring good results and meet the expectations of others, they will have a strong willingness to participate. In terms of altruism, it mainly depends on the level of their personally embedded moral standards. Whether morality is stimulated or not depends on the public's awareness of the results of participation or non-participation and whether they attribute these results to themselves. When the public is aware of the benefits of participating in RLG, such as improving the ecological environment, and think that they are responsible for the negative results of non-participation, they will develop a strong intention to participate. The study combined different influencing factors to strengthen the understanding of public participation in RLG.

The research showed that in the context of self-interest and altruism, the hypothesis was still tested, but the higher explanatory power could better explain that the behavior of public participation in RLG is not affected by a single variable but by the joint action of

self-interest (individual rationality) and altruism (social rationality). From the perspective of individual rationality, the intention of the public to participate in RLG had a significant positive impact on behavior, and behavioral attitudes and subjective norms could better predict the intention to participate. However, their roles were different, and subjective norms had a greater impact on the intention to participate. From the perspective of social rationality, awareness of negative consequences positively affects personal norms through AP, and positive PN (moral perception) promotes public participation in RLG. In addition, GN is another important factor driving the public to participate in RLG; GN directly activates public participation in RLG.

The findings are of important theoretical significance for further exploring the origin of public willingness to participate in RLG, which is mainly reflected in the following: ① all variables in TPB and NAM were included in the analysis, which overcomes the limitations of a single theoretical analysis framework to a certain extent and provides a reference for the subsequent use of the TPB-NAM integrated framework to analyze public pro-environmental behavior; ② it demonstrated the effectiveness of combining government norms with the TPB and provided a valuable attempt to further expand the TPB-NAM integrated analysis framework; and ③ based on the two perspectives of self-interest (individual rationality) and altruism (social rationality), it was demonstrated that in addition to self-interest motivation, moral factors are also an important component that influences public participation in RLG, which provides a multi-dimensional perspective for further studying public participation in RLG.

*5.2. Suggestions*

(1) The government should take corresponding measures to cultivate a positive attitude of public participation in RLG, such as publicizing the economic and ecological benefits of RLG through television, radio, Wechat groups, and agriculture-related network platforms. It should also organize activities such as visiting sample rivers and lakes to deepen the public's perception and experience of RLG, so as to effectively increase the public's intention to govern rivers and lakes. River chiefs at all levels should actively strengthen positive publicity, create a positive atmosphere to regulate public participation in RLG, make full use of the social relationship network of rural acquaintances along the river and lake, and promote public participation in RLG through the guidance and demonstration of important figures such as relatives, friends, villagers, and village leaders so as to stimulate the subjective norms of public participation in RLG and enhance the understanding of public participation in river and lake protection and governance.

(2) The river chiefs should raise public awareness of adverse results of the decline in the environmental quality of rivers and lakes by holding meetings, delivering theme lectures, distributing promotional manuals, or carrying out multimedia publicity activities. Cognitive drive is an important driving force for the public's voluntary participation. They should coordinate the interests of political and social governance, strengthen the publicity of the river governor system, improve the public's awareness of participating in the river governor system in water control through multiple channels, reinforce the efficacy of public participation, enhance its institutional trust, and promote public participation in water control as a conscious action.

(3) The behavior of the government in dealing with public participation in RCS should be normalized to improve the credibility of the government. The relevant departments should ensure the openness and responsiveness of the government's behavior, handle the supervision of the public in a timely manner, and make public the policy actions of RLG, the effects of governance, river basin environmental data, and the results of supervision and feedback that the public are concerned about. Through suggestions, rewards, honorary recognition, financial subsidies, and other measures, they should improve the sense of honor and gain of public participation in water control, let the



public truly find the benefits of water participation, improve the effective awareness of public participation, and stimulate public enthusiasm for participation.

*5.3. Limitations and Future Research*

Although this paper provided some insights on the intention and behavior of public participation in RLG in the context of the RCS, there were also some limitations. First of all, the sample size in this study was relatively small, and only individuals in Henan Province of China were used as the research sample, so there are certain limitations in the survey data. In the RCS context, the effect of local governments encouraging the public to participate in RLG is quite different, so in the future, we will further conduct the relevant study in other populations and regions in China. Second, the estimated models in this paper only evaluated the intention and behavior of the public to participate in RLG in some areas. Future research can develop a more rigorous sampling plan, test the estimated models in more representative areas, and compare it with the current research results. Finally, although this study successfully integrated TPB and NAM into the hypothetical model, the decision making of public participation in RLG is more complex than the theoretical models. Public participation in RLG is the result of the joint action and guidance of individual psychological, external environment, and other factors. In the future, we should further explore the driving mechanism of other variables (such as perceived risk) for the intention and behavior of public participation in RLG and explore the influencing factors that cause the contradiction between the two aspects, which will help improve the conversion efficiency from intention to behavior.

**Author Contributions:** Conceptualization, X.Z. and L.L.; methodology, X.Z.; software, X.Z. and H.L.; validation, X.Z. and H.L.; formal analysis, X.Z.; investigation, X.Z. and H.L.; resources, Z.S. and X.L.; data curation, Z.S. and L.L.; writing–original draft preparation, X.Z.; writing–review and editing, X.Z.; visualization, L.L.; supervision, X.Z.; project administration, Z.S.; funding acquisition, X.Z. All authors have read and agreed to the published version of the manuscript.

**Funding:** This work was financially supported by National Natural Science Foundation of China (51979108) and Key Science and Technology Project of China National Tobacco Corporation Henan Branch (2020410000270020).

**Data Availability Statement:** The datasets used and/or analyzed during the current study are available from the corresponding author on reasonable request.

**Conflicts of Interest:** The authors declare no conflict of interest.

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
