# Peer review of "Study on Factors Influencing Public Participation in River and Lake Governance in the Context of the River Chief System—Based on the Integrated Model of TPB-NAM"

_water, doi:10.3390/w15020275_

Round 1
Reviewer 1 Report
Water 2112277 review
Study on Influencing Factors of Public Participation in the River and Lake Governance….
Overall, this is a well written research paper, but I have several suggestions for improvement of the paper. It will be good to occasionally spell out the many acronyms used in the paper to save the reader from going back and forth at key points of discussion
Introduction
Please occasionally spell out RLG and RCS throughout the paper.
Line 58 -suggest -more effective….
Theoretical framework and research hypothesis
Line 113- suggest- the more likely public…
Line 147- please spell out NAM
Line 191- suggest -the less likely residents…
Line 202- please spell out TPB
Results
Line 237 -how were the respondents randomly distributed?
Line 241- how were the experts in the field selected?
Discussion
Pleaser spell out acronyms occasionally
Line 330- spell out NFI RMSEA
Conclusions
Line 412- suggest- will be motivated by strong intentions to participant…
Authors should discuss the generalizability of the study results to other populations and regions in China
Author Response
Thanks for the reviewers’ valuable comments. After carefully considering the reviewers' comments and suggestions, the authors have made substantive and serious modifications to the paper, which can be seen in the following aspects for details of the modifications:
Point 1: Please occasionally spell out RLG and RCS throughout the paper.Line 58 -suggest -more effective….
Response 1: It has been modified according to the suggestions of the reviewer
Point 2: Theoretical framework and research hypothesis.Line 113- suggest- the more likely public…Line 147- please spell out NAM.Line 191- suggest -the less likely residents….Line 202- please spell out TPB
Response 2: According to the reviewer proposal related to acronyms, we have modified the expressions which the reviewer pointed out.
Point 3: Line 237 -how were the respondents randomly distributed? Line 241- how were the experts in the field selected?
Response 3: According to the opinion proposed by the reviewer, Line 237 -how were the respondents randomly distributed? Line 241- how were the experts in the field selected? We have added the expressions “The survey was conducted in 18 cities under the jurisdiction of Henan Province. 18 cities served as sample areas, 3 or 4 towns were randomly selected according to the population and economic conditions of each sample city, and 3 or 4 villages were randomly selected from each township, which fully reflects the representativeness of the samples in terms of geographical distribution and socio-economic development level.” and “The specialists include 12 experts and scholars, who are mainly engaged in RLG and RCS in universities and scientific research institutes, and 8 government staff, who are mainly working staff of the River Lake Protection Center of the Ministry of Water Re-sources of China.”
Point 4:Pleaser spell out acronyms occasionally. Line 330- spell out NFI RMSEA
Response 4: According to the reviewer proposal related to acronyms, we have modified the expressions which the reviewer pointed out.
Point 5: Line 412- suggest- will be motivated by strong intentions to participant…Authors should discuss the generalizability of the study results to other populations and regions in China
Response 5: According to the third opinion proposed by the reviewer that authors should discuss the generalizability of the study results to other populations and regions in China, we added the expression “the sample size in this study is relatively small, and only the individuals in Henan Province of China have been taken as the research sample, so there are certain limitations in the survey data. In RCS context, the effect of local governments promoting the public to participate in RLG is quite different, so in future we will further conduct the relevant study in other populations and regions in China”.

Reviewer 2 Report
The article aims to explore the influencing factors and driving mechanisms of public participation in the river and lake governance in the context of the river chief system from aspects of self-interest and altruism. The article obtained 508 sample data from a survey of residents and analyzed the results through structural equation modelling. The results of the article give good suggestions for public participation under the river chief system, But I personally think that the following major problems need to be solved first,
Comment 1:In the section ‘Abstract’, The authors set this out in text in the summary of results and I suggest that the authors enrich the data description to support the results.
Comment 2:In the section‘1. Introduction’, The author does not state what the marginal contribution of this article is, and I suggest that the author add how the article differs from established articles and what the innovations are.
Comment 3: In the section ‘3.1. Scale design’, The authors design questions for each indicator, but there is repetition and poor logic between questions. I would suggest that the authors sort out the logic of the questions and add elaboration to make the indicators more scientific.
Comment 4:In the section ‘4.3 Competing model 3’, The authors do not give explanations for many of the results in their analysis and I suggest that they explore the reasons for these results and also compare them to other literature
Comment 5: After analysing the results of the three models individually, the authors do not compare the similarities and differences in the results of the three models, and I suggest that the authors add a comparative analysis of the results of the three models.
Comment 6: In the section ‘1 Introduction’, I suggest that the authors add citations related to river governance.
Reviews on land use change induced effects on regional hydrological ecosystem services for integrated water resources management[J]. Physics and Chemistry of the Earth, Parts A/B/C, 2015.
Water governance and scalar politics across multiple-boundary river basins: states, catchments and regional powers in the Iberian Peninsula[J]. Water International, 2014, 39(3):333-347.
Author Response
Thanks for the reviewers’ valuable comments. After carefully considering the reviewers' comments and suggestions, the authors have made substantive and serious modifications to the paper, which can be seen in the following aspects for details of the modifications:
Point 1: In the section ‘Abstract’, The authors set this out in text in the summary of results and I suggest that the authors enrich the data description to support the results.
Response 1: According to the reviewer's first revision proposal, this paper further enrich the data description to support the results in the abstract, and adds the expression " The explanatory power of TPB-NAM model (R2=60.7%) is superior to that of the extended TPB model (R2=60.0%) and NAM model (R2=50.0%)."
Point 2: In the section‘1. Introduction’, The author does not state what the marginal contribution of this article is, and I suggest that the author add how the article differs from established articles and what the innovations are.
Response 2: According to the reviewer's second proposal, we have added the marginal contribution of this article in the Introduction,which is expressed as “The marginal contribution of this paper is that it combines TPB with NAM, and applies the integrated model of TPB-NAM to the study of public participation in RLG in RCS context, which provides a new perspective for the study of public pro-environmental behavior.”
Point 3: In the section ‘3.1. Scale design’, The authors design questions for each indicator, but there is repetition and poor logic between questions. I would suggest that the authors sort out the logic of the questions and add elaboration to make the indicators more scientific.
Response 3: According to the reviewer's third revision suggestion, we have checked and sorted out the index design, the specific content is shown in Table 2, and the relevant values in the results have also been updated.
Point 4: In the section "4.3 Competitive Model 3", the authors do not explain many of the results in their analysis, and I suggest that they explore the reasons for these results and compare them with other literature.
Response 4: Based on the reviewer's fourth revision suggestion, we supplemented the results of competition Model 3 and then compared them with other literature.
Viewpoint 5: After analyzing the results of the three models respectively, I suggest that the author make a comparative analysis of the results of the three models without comparing the similarities and differences.
Response 5: According to the reviewer's fifth revision suggestion, we supplemented the discussion in 4.2 and made a comparative analysis of the results of the three models.
Viewpoint 6:In the section "1 Introduction", I suggest that the author add citations related to river governance.
Response 6: In accordance with the reviewer's sixth revision proposal, we have added two citations related to river governance to the introduction and references.

Round 2
Reviewer 2 Report
none